# Softened Symbol Grounding for Neuro-symbolic Systems

**Zenan Li[1], Yuan Yao[1], Taolue Chen[2], Jingwei Xu[1], Chun Cao[1], Xiaoxing Ma[1], Jian Lü[1]**

[1]State Key Lab of Novel Software Technology, Nanjing University, China
[2]Department of Computer Science, Birkbeck, University of London, UK
`lizn@smail.nju.edu.cn`, `t.chen@bbk.ac.uk`,
`{y.yao,jingweix,caochun,xxm,lj}@nju.edu.cn`

## ABSTRACT

Neuro-symbolic learning generally consists of two separated worlds, i.e., neural network training and symbolic constraint solving, whose success hinges on symbol grounding, a fundamental problem in AI. This paper presents a novel, softened symbol grounding process, bridging the gap between the two worlds, and resulting in an effective and efficient neuro-symbolic learning framework. Technically, the framework features (1) modeling of symbol solution states as a Boltzmann distribution, which avoids expensive state searching and facilitates mutually beneficial interactions between network training and symbolic reasoning; (2) a new MCMC technique leveraging projection and SMT solvers, which efficiently samples from disconnected symbol solution spaces; (3) an annealing mechanism that can escape from sub-optimal symbol groundings. Experiments with three representative neuro-symbolic learning tasks demonstrate that, owing to its superior symbol grounding capability, our framework successfully solves problems well beyond the frontier of the existing proposals.

## 1 INTRODUCTION

Neuro-symbolic systems have been proposed to connect neural network learning and symbolic constraint satisfaction (Garcez et al., 2019; Marra et al., 2021; Yu et al., 2021; Hitzler, 2022). In these systems, the neural network component first recognizes the raw input as a symbol, which is further fed into the symbolic component to produce the final output (Yi et al., 2018; Li et al., 2020; Liang et al., 2017). Such a neuro-symbolic paradigm has shown unprecedented capability and achieved impressive results in many tasks including visual question answering (Yi et al., 2018; Vedantam et al., 2019; Amizadeh et al., 2020), vision-language navigation (Anderson et al., 2018; Fried et al., 2018), and math word problem solving (Hong et al., 2021; Qin et al., 2021), to name a few.

As exemplified by Figure 1, to maximize generalizability, such problems are usually cast in a weakly-supervised setting (Garcez et al., 2022): the final output of the neuro-symbolic computation is provided as supervision during training rather than the label of intermediate symbols. Lacking direct supervised labels for network training appeals for an effective and efficient approach to solve the *symbol grounding* problem, i.e., establishing a feasible and generalizable mapping from the raw inputs to the latent symbols. Note that bypassing symbol grounding (by, e.g., regarding the problem as learning with logic constraints) is possible, but cannot achieve a satisfactory performance (Manhaeve et al., 2018; Xu et al., 2018; Pryor et al., 2022). Existing methods incorporating symbol grounding in network learning heavily rely on a good initial model and perform poorly when starting from scratch (Dai et al., 2019; Li et al., 2020; Huang et al., 2021).

A key challenge of symbol grounding lies in the semantic gap between neural learning which is stochastic and continuous, and symbolic reasoning which is deterministic and discrete. To bridge the gap, we propose to *soften* the symbol grounding. That is, instead of directly searching for a deterministic input-symbol mapping, we optimize their Boltzmann distribution, with an annealing strategy to gradually converge to the deterministic one. Intuitively, the softened Boltzmann distribution provides a playground where the search of input-symbol mappings can be guided by the neural

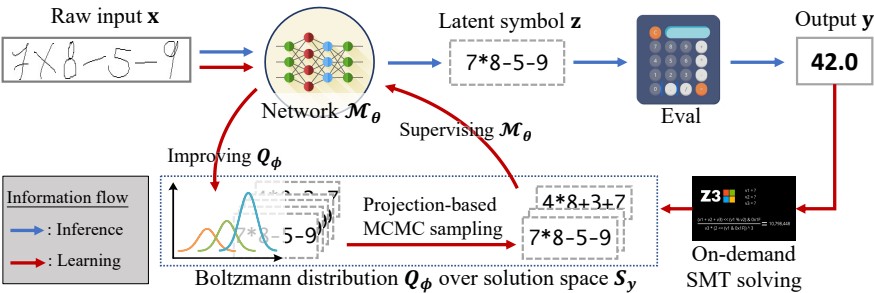

Figure 1: An example neural-symbolic system for handwritten formula evaluation. It takes a handwritten arithmetic expression $\mathbf{x}$ as input and evaluates the expression to output $\mathbf{y}$. The neural network component $\mathcal{M}_{\boldsymbol{\theta}}$ recognizes the symbols $\mathbf{z}$ (i.e., digits and operators) in the expression, and the symbolic component evaluates the recognized formula by, e.g., the Python function 'eval'. The challenge in training $\mathcal{M}_{\boldsymbol{\theta}}$ comes from the lack of explicit $\mathbf{z}$ to bridge the gap between the neural world ($\mathbf{x}$ to $\mathbf{z}$) and the symbol world ($\mathbf{z}$ to $\mathbf{y}$). Through softened symbol grounding, the model training and the constraint satisfaction join force to resolve the latent $\mathbf{z}$ to fit both the given $\mathbf{x}$ and $\mathbf{y}$.

network, and the network training can be supervised by sampling from the distribution. Game theory indeed provides a theoretical support for this strategy (Conitzer, 2016): the softening makes the learning process a series of mixed-strategy games during the annealing process, which encourages stronger interactions between the neural and symbolic worlds.

The remaining challenge is how to efficiently sample the feasible input-symbol mappings. Specifically, feasible solutions are extremely sparse in the entire symbol space and different solutions are poorly connected, which prevents the Markov Chain Monte Carlo (MCMC) sampling from efficiently exploring the solution space. To overcome this deficiency, we leverage the projection technique to accelerate the random walk for sampling (Feng et al., 2021b), aided by satisfiability modulo theory (SMT) solvers (Nieuwenhuis et al., 2006; Moura & Bjørner, 2008). The intuition is that disconnected solutions in a high-dimensional space may become connected when they are projected onto a low-dimensional space, resulting in a rapid mixing time of the MCMC sampling (Feng et al., 2021a). The SMT solver, which is called on demand, is used as a generic approach to compute the inverse projection. Although MCMC sampling and SMT solvers may introduce bias, the theoretical result confirms that it can be pleasantly offset by the proposed stochastic gradient descent algorithm.

## 2 SOFTENING SYMBOL GROUNDING

Throughout this paper, we refer to $\mathcal{X}$ as the input space of the neuro-symbolic system, and $\mathcal{Z}$ as its *symbol space* or *state space* (e.g., all legal and illegal arithmetic expressions in the HWF task). We consider the neuro-symbolic computing task which first trains a neural network (parameterized by $\boldsymbol{\theta}$), mapping a raw input $\mathbf{x} \in \mathcal{X}$ to some latent state $\mathbf{z} \in \mathcal{Z}$ with a (variational) probability distribution $P_{\boldsymbol{\theta}}(\mathbf{z}|\mathbf{x})$. The state $\mathbf{z}$ is further fed into a predefined symbolic reasoning procedure to produce the final output $\mathbf{y}$. The training data contains only the input $\mathbf{x}$'s and the corresponding $\mathbf{y}$'s, which casts the problem into the so-called weakly-supervised setting. In general, we formulate the pre-defined symbolic reasoning procedure and the output $\mathbf{y}$ as a set of *symbolic constraint* $\mathcal{S}_{\mathbf{y}}$ on the symbol space. For instance, in Figure 1, the constraint specifies that the arithmetic expressions must evaluate to 42. We say a state $\mathbf{z}$ is *feasible* or satisfies the symbolic constraint if $\mathbf{z} \in \mathcal{S}_{\mathbf{y}}$.

The major challenge in this neuro-symbolic learning paradigm lies in the *symbol grounding problem*, i.e., establishing a mapping $h: \mathcal{X} \to \mathcal{Z}$ from the raw input to a feasible state that satisfies the symbolic constraint. Specifically, an effective mapping $h$ should enable the model to explain as many observations as possible. As a result, the symbol grounding problem on a given dataset $\{(\mathbf{x}^i, \mathbf{y}^i)\}_{i=1,\dots,N}$ can be formulated as

$$\min_{h} \left\{ \min_{\boldsymbol{\theta}} \ell(\boldsymbol{\theta}) := -\sum_{i=1}^{N} \log P_{\boldsymbol{\theta}}(\mathbf{z}^i|\mathbf{x}^i) \right\} \quad \text{s.t.} \quad \mathbf{z}^i = h(\mathbf{x}^i) \in \mathcal{S}_{\mathbf{y}^i}, i = 1, \dots, N. \quad (1)$$

A straightforward solution to the above formulation would first train a network for each feasible mapping, and then select the one that achieves the final output $\mathbf{y}$ with a maximum likelihood. However, this solution is impractical since the number of feasible mappings grows exponentially.

An obvious shortcoming of the above solution is that the neural network learning process makes no use of the knowledge embodied in the symbolic constraint. Vice versa, searching for the best mapping is not guided by the network. To overcome this shortcoming, one can switch the minimization order in problem (1), and obtain a new but numerically equivalent problem:

$$\min_{\boldsymbol{\theta}} \Big\{ \min_{h} \ell(\boldsymbol{\theta}) := -\sum_{i=1}^{N} \log P_{\boldsymbol{\theta}}(\mathbf{z}^i|\mathbf{x}^i) \quad \text{s.t.} \quad \mathbf{z}^i = h(\mathbf{x}^i) \in \mathcal{S}_{\mathbf{y}^i}, i = 1, \ldots, N \Big\}. \tag{2}$$

The optimization problem (2) first determines a "best" mapping based on the initial model, and then updates the model to fit this mapping. The two steps are iterated until no improvement can be made. However, this grounding strategy may easily get trapped into a local optimum. The reason is that, every time a feasible mapping $h$ is achieved, $h$ tends to direct the neural network to (over)fit itself. Because the mappings are deterministic and discrete, there is no smooth route to alternative feasible mappings that would further improve the network. This insufficient information exchange between network training and symbolic reasoning makes the success of symbol grounding highly dependent on the quality of the initial model.

In this work, we propose to soften the symbol grounding to facilitate the interaction between neural perception and symbolic reasoning. Instead of directly searching for a deterministic mapping $h$, we first pursue an optimal probability distribution of $h$, and then gradually "sharpen" the distribution to obtain the final deterministic $h$. Formally, for each input $\mathbf{x}$, we introduce a Boltzmann distribution $Q_{\boldsymbol{\phi}}$ over $\mathcal{S}_{\mathbf{y}}$, parameterized by $\boldsymbol{\phi}$, to indicate the probability of each feasible state that satisfies the symbolic constraint. Then, the softened symbol grounding problem can be formulated as follows:

$$\min_{\boldsymbol{\theta}, \boldsymbol{\phi}} \quad \ell(\boldsymbol{\theta}, \boldsymbol{\phi}) := -\sum_{i=1}^{N} \sum_{\mathbf{z}^i \in \mathcal{S}_{\mathbf{y}^i}} Q_{\boldsymbol{\phi}}^i(\mathbf{z}^i) \log P_{\boldsymbol{\theta}}(\mathbf{z}^i|\mathbf{x}^i) + \gamma Q_{\boldsymbol{\phi}}^i(\mathbf{z}^i) \log Q_{\boldsymbol{\phi}}^i(\mathbf{z}^i) \tag{P}$$

$$\text{s.t.} \quad \operatorname{supp}(Q_{\boldsymbol{\phi}}^i) \subseteq \mathcal{S}_{\mathbf{y}^i}, \quad i = 1, \ldots, N,$$

where $\operatorname{supp}(Q_{\boldsymbol{\phi}})$ denotes the support of $Q_{\boldsymbol{\phi}}$. The entropy term $\sum_{\mathbf{z} \in \mathcal{S}_{\mathbf{y}}} Q_{\boldsymbol{\phi}}(\mathbf{z}) \log Q_{\boldsymbol{\phi}}(\mathbf{z})$ is introduced to control the sharpness of $Q_{\boldsymbol{\phi}}$ (MacKay et al., 2003), and the decreasing of its coefficient $\gamma$ ensures that the grounding can converge to a deterministic mapping $h$. Except for the case $\gamma = 1$ which yields the KL divergence between $P_{\boldsymbol{\theta}}$ and $Q_{\boldsymbol{\phi}}$, we also examine two extreme cases: (1) when $\gamma \to +\infty$, $Q_{\boldsymbol{\phi}}$ is forced towards the uniform distribution, and thus the minimization only aims to restrict the support of $P_{\boldsymbol{\theta}}$ to $\mathcal{S}_{\mathbf{y}}$; (2) when $\gamma \to 0$, $Q_{\boldsymbol{\phi}}$ is confined to a one-hot categorical distribution, reducing to directly search for the deterministic mapping $h$.

**Advantages**. Game theory provides a perspective to understand why our softened strategy improves over problems (1) and (2) with better interaction between model training and symbolic reasoning. Either problem (1) or (2) can be viewed as a pure-strategy Stackelberg game. That is, both the model training and the symbolic reasoning are forced to take a certain action (e.g., selecting a deterministic mapping $h$) during optimization. In contrast, problem (P) can be seen as a Stackelberg game with mixed strategies, where the player takes a randomized action with the distribution $Q_{\boldsymbol{\phi}}$. Compared with the pure strategy, a mixed strategy does provide more information of the game, and thus strictly improves the utility of model training (Letchford et al., 2014; Conitzer, 2016).

In addition, this softening technique also avoids enumeration and thus improves efficiency. In problem (1), for each input $\mathbf{x}$, the minimization in its corresponding symbol $\mathbf{z}$ needs searching over the whole $\mathcal{S}_{\mathbf{y}}$ (i.e., enumerating all feasible states satisfying the symbolic constraint). This is generally an intractable *#P-complete* problem in theory (Arora & Barak, 2009). Problem (P) circumvents this costly computation by estimating the expectation over $Q_{\boldsymbol{\phi}}$, which can be efficiently computed with a tailored sampling strategy discussed in the next section.

## 3 Markov chain Monte Carlo Estimate via Projection

To simplify presentation, in this section, we consider a single data sample. Specifically, by removing the summation over all samples and dropping the superscripts, problem (P) can be formulated as

$$\min_{\boldsymbol{\theta},\boldsymbol{\phi}} \ell(\boldsymbol{\theta},\boldsymbol{\phi}) := \sum_{\mathbf{z}\in\mathcal{S}_{\mathbf{y}}} Q_{\boldsymbol{\phi}}(\mathbf{z})\log P_{\boldsymbol{\theta}}(\mathbf{z}|\mathbf{x}) + \gamma Q_{\boldsymbol{\phi}}(\mathbf{z})\log Q_{\boldsymbol{\phi}}(\mathbf{z}), \quad \mathrm{supp}(Q_{\boldsymbol{\phi}}) \subseteq \mathcal{S}_{\mathbf{y}}. \tag{3}$$

This problem can be solved by alternating between the gradient descent step in $\boldsymbol{\theta}$ and the minimization step in $\boldsymbol{\phi}$. The updates of $\boldsymbol{\theta}$ and $\boldsymbol{\phi}$ at the $k$-th iteration are

$$\boldsymbol{\theta}_{k+1} = \boldsymbol{\theta}_k - \eta\nabla_{\boldsymbol{\theta}}\ell(\boldsymbol{\theta}_k,\boldsymbol{\phi}_k), \quad \boldsymbol{\phi}_{k+1} = \operatorname*{arg\,min}_{\boldsymbol{\phi}|\mathrm{supp}(Q_{\boldsymbol{\phi}})} \ell(\boldsymbol{\theta}_{k+1},\boldsymbol{\phi}). \tag{4}$$

Note that the closed-form solution $Q_{\boldsymbol{\phi}^*}$ exists when $P_{\boldsymbol{\theta}}$ is fixed, ensuring the convergence of gradient descent in $\boldsymbol{\theta}$ (Jin et al., 2020, Theorem 31). For details, the lower-level problem, i.e., $\min_{\boldsymbol{\phi}}\ell(\boldsymbol{\theta},\boldsymbol{\phi})$, is strictly convex, and thus contains the unique minimum:

$$Q_{\boldsymbol{\phi}^*}(\mathbf{z}) = \begin{cases} P_{\boldsymbol{\theta}}(\mathbf{z}|\mathbf{x})^{\frac{1}{\gamma}}/\sum_{\mathbf{z}'\in\mathcal{S}_{\mathbf{y}}} P_{\boldsymbol{\theta}}(\mathbf{z}'|\mathbf{x})^{\frac{1}{\gamma}}, & \text{if } \mathbf{z}\in\mathcal{S}_{\mathbf{y}}, \\ 0, & \text{otherwise.} \end{cases} \tag{5}$$

Given the closed-form solution $Q_{\boldsymbol{\phi}^*}$, the loss function $\ell(\boldsymbol{\theta},\boldsymbol{\phi}^*)$ and its gradient $\nabla_{\boldsymbol{\theta}}\ell(\boldsymbol{\theta},\boldsymbol{\phi}^*)$ can be estimated through Monte Carlo sampling on $Q_{\boldsymbol{\phi}^*}$.

The remaining problem is how to sample $Q_{\boldsymbol{\phi}^*}$, which is challenging due to the unknown structure of $\mathcal{S}_{\mathbf{y}}$. Existing methods usually sample from the entire symbol/state space $\mathcal{Z}$, and then either reject the state $\mathbf{z}\notin\mathcal{S}_{\mathbf{y}}$ (e.g., policy-gradient method (Williams, 1992)), or project the infeasible state $\mathbf{z}$ to $\mathcal{S}_{\mathbf{y}}$ (e.g., back-search method (Li et al., 2020)). Unfortunately, these methods suffer from the *sparsity problem*, i.e., feasible $\mathbf{z}$'s are very sparse in $\mathcal{Z}$, causing the policy-gradient to vanish and the back-search to fail.

To overcome the sparsity problem, we propose to directly sample from the symbolic constraint $\mathcal{S}_{\mathbf{y}}$ (i.e., the solution space). By applying the Metropolis algorithm (Bhanot, 1988; Beichl & Sullivan, 2000), the acceptance ratio of jumping from one feasible state $\mathbf{z}$ to another one $\mathbf{z}'$ does not vanish, and can be computed as

$$\tau = \frac{Q_{\boldsymbol{\phi}^*}(\mathbf{z}')}{Q_{\boldsymbol{\phi}^*}(\mathbf{z})} = \left(\frac{P_{\boldsymbol{\theta}}(\mathbf{z}'|\mathbf{x})}{P_{\boldsymbol{\theta}}(\mathbf{z}|\mathbf{x})}\right)^{\frac{1}{\gamma}}. \tag{6}$$

Hence the problem becomes: (1) how to generate an initial state $\mathbf{z}$, and (2) how to jump from $\mathbf{z}$ to $\mathbf{z}'$. For the former, a natural way is to leverage SMT solvers (Moura & Bjørner, 2008).[1] For the latter, the most commonly used strategy is to achieve the new state via random walk (Sherlock et al., 2010). However, there lacks a systematic random walk approach in the solution space, because the solution space is likely unconnected (Wigderson, 2019), creating the so-called *connectivity barrier*.

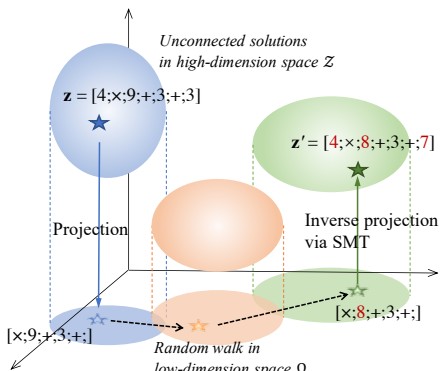

Figure 2: Sampling unconnected solutions via projection. For our running example, we use the projection $\Pi([\mathbf{z}_1;\dots;\mathbf{z}_7]) = [\mathbf{z}_2;\dots;\mathbf{z}_6]$, i.e., dropping the first and the last digits. The current state $\mathbf{z} = [4;\times;9;+;3;+;3]$ is projected to $\Pi(\mathbf{z}) = [\times;9;+;3;+;]$. We then randomly select an individual component, say, '9', and update it to '8'. Next, a new feasible state (e.g., $4\times8+3+7=42$) is derived by computing the inverse projection of $\Pi(\mathbf{z}') = [\times;8;+;3;+;]$ with an SMT solver.

Inspired by Feng et al. (2021a), we propose to overcome the connectivity barrier by the *projection* technique. Elaborately, we introduce a projection operator $\Pi(\cdot)\colon \mathcal{Z}\to\Omega$ that maps the state space $\mathcal{Z}$ to a lower-dimension space $\Omega$, and then apply the single-site Metropolis algorithm in $\Omega$. The projection essentially compacts the state space, and thus significantly improves the connectivity of the solution space. Figure 2 illustrates the key idea. Consider the running example in Figure 1, where $\mathcal{S}_{\mathbf{y}}$ requires that all expressions are evaluated to 42. The SMT solver, together with the standard single-site Metropolis (a.k.a. Metropolis-in-Gibbs) (Metropolis et al., 1953; Bai, 2009),

---

[1]Current SMT solvers are mainly designed for the satisfaction problem, namely, they are efficient in finding a solution, but underperform in generating all solutions.

can easily derive an initial state (e.g., $4 \times 9 + 3 + 3 = 42$) satisfying the symbolic constraint, but cannot further explore other feasible states due to the connectivity barrier (Ermon et al., 2012).[2] In contrast, in the lower-dimension space $\Omega$, it is much easier to jump to another feasible state.

## 4 ALGORITHM AND ANALYSIS

The overall algorithm of our neuro-symbolic learning is shown in Alg. 1. In a nutshell, we first conduct a few random walk steps to sample a new state $\mathbf{z}$ on the distribution $Q_{\phi^*}$; we then estimate the gradient based on $\mathbf{z}$, and conduct one stochastic gradient descent step. As shown by Feng et al. (2021a), under some proper assumptions, the Metropolis algorithm enjoys the *rapid mixing* property on the projection space (Levin & Peres, 2017). Therefore, we can efficiently construct the approximate sampling on $Q_{\phi^*}$, without taking too many steps in the Metropolis algorithm. Additionally, both the sampling method and the SMT solver can be paralleled for different examples, hence the batch gradient descent is well supported in our framework.

---

**Algorithm 1** Neural Symbolic Learning Procedure

---
Set an initial value of $\gamma$.
Calculate an initial state $\mathbf{z} \in \mathcal{S}_\mathbf{y}$ for each input-output pair $(\mathbf{x}, \mathbf{y})$ via the SMT solver.
**for** $k = 0, 1, \ldots, K$ **do**
    Randomly draw an example $(\mathbf{x}, \mathbf{y})$ from training data $\{(\mathbf{x}_i, \mathbf{y}_i)\}_{i=1}^N$.
    **for** $t = 0, 1, \ldots, T$ **do**
        *Generate new state*
        Compute the projection $\boldsymbol{u} = \Pi(\mathbf{z}) \in \Omega$.
        Obtain $\boldsymbol{u}'$ by randomly selecting and updating a component of $\boldsymbol{u}$.
        Calculate the inverse projection $\mathbf{z}' = \Pi^{-1}(\boldsymbol{u}')$ via the SMT solver, as a new state.
        *Accept/reject new state*
        Calculate the acceptance ratio $\tau = (P_{\boldsymbol{\theta}}(\mathbf{z}'|\mathbf{x})/P_{\boldsymbol{\theta}}(\mathbf{z}|\mathbf{x}))^{1/\gamma}$.
        Generate a uniform random number $\nu \in [0, 1]$.
        Update the state $\mathbf{z}$ to $\mathbf{z}'$ if $\nu \le \tau$.
    **end for**
    *Train the network*
    Estimate the gradient $\hat{\nabla}\ell(\boldsymbol{\theta}) = -\nabla_{\boldsymbol{\theta}} \log P_{\boldsymbol{\theta}}(\mathbf{z}|\mathbf{x})$.
    Update network parameters $\boldsymbol{\theta}$ by the stochastic gradient decent.
    Decrease the coefficient $\gamma$.
**end for**

---

**Convergence.** In the ideal case, the gradient estimate $\hat{\nabla}\ell(\boldsymbol{\theta})$ is unbiased, and the gradient descent in $\boldsymbol{\theta}$ (i.e., the network parameters) can converge. However, the bias is introduced due to: 1) the approximate sampling of Metropolis algorithm (Jacob et al., 2017); 2) the inverse projection implemented by the SMT solver (Moura & Bjørner, 2008). For the former, we have to increase the number of inner iterations in our algorithm or consider adaptive variants of the Metropolis algorithm. For the latter, we can alter the projection operator during the training process, or increase the dimension of projection space. Nevertheless, none of these methods can fully avoid the bias of gradient estimate. To this end, we provide a convergence result for the stochastic gradient descent with limited bias.

**Proposition 1.** *Assume the loss function $\ell(\boldsymbol{\theta})$ is $L$-Lipschitz and $\ell$-smooth, and let the actual sampling distribution be $\widehat{Q}$. Then, if the total variation distance $d_{tv}(\widehat{Q}, Q^*)$ is bounded by $\epsilon$, it holds after $K$ steps of the stochastic gradient descent with learning rate $\eta = \alpha/(\sqrt{T+1})$:*

$$\frac{1}{K} \sum_{k=1}^{K} \|\nabla\ell(\boldsymbol{\theta}_k)\|^2 \le \mathcal{O}\left(\frac{\ell\sigma^2 + \Delta_0}{\alpha\sqrt{T+1}}\right) + (n\epsilon L)^2,$$

*where $\Delta_0 = \ell(\boldsymbol{\theta}_0) - \min \ell(\boldsymbol{\theta})$, and $n$ is the cardinal number of $\mathrm{supp}(Q^*)$.*

---

[2]To maintain stability, the single-site Metropolis conducts random walk in a component-wise way. That is, in each iteration, it randomly selects and updates an individual component in the current state $\mathbf{z}$ to generate a new state $\mathbf{z}'$. However, it can be observed that the update of any individual component (i.e., '4','×', '9', '+', '3', '+', '3') will result in no feasible new states.

*Remarks.* A proof is given in Appendix A.1. This proposition states that the stochastic gradient descent with MCMC gradient estimate converges to an approximate stationary point. Moreover, the bias term is gradually wiped out in the training process, since the decreasing of $\gamma$ shrinks the support of $Q^*$, making the gradient estimate finally align with the true one.

**Generalization of existing methods.** Existing neuro-symbolic learning frameworks, viz. semantic loss (Xu et al., 2018), deepproblog (Manhaeve et al., 2018), and neural-grammar-symbolic learning (Li et al., 2020), can be understood as special cases of our framework.

**Proposition 2.** *All three frameworks (semantic loss, deepproblog, and neural-grammar-symbolic learning) share the same loss function*

$$\hat{\ell}(\boldsymbol{\theta}) := -\sum_{i=1}^{N} \sum_{\mathbf{z}^i \in \mathcal{S}_{\mathbf{y}}^i} \log P_{\boldsymbol{\theta}}(\mathbf{z}^i | \mathbf{x}^i),$$

*and they are equivalent to Problem (P) with a fixed $\gamma$ ($\gamma = 1$). Here, the equivalence means that the problems have the same optimal solution and gradient descent dynamics.*

*Remarks.* The proof is in Appendix A.2. Compared with minimizing $\hat{\ell}$, our framework enjoys two advantages: (i) the Boltzmann distribution $Q$ is explicitly expressed, making the sampling tractable and easy to implement even when the state space is very large; (ii) the annealing strategy of $\gamma$ largely alleviates the sensitivity to the initial point, guiding to a better optimal solution.

**Annealing strategy.** Next, we discuss the decreasing strategy of $\gamma$. By setting $\phi_{\mathbf{z}}^* = -\log P_{\boldsymbol{\theta}}(\mathbf{z}|\mathbf{x})$ as the entropy for each state $\mathbf{z}$ (Thomas & Joy, 2006), we can obtain that

$$Q_{\boldsymbol{\phi}^*}(\mathbf{z}) = \frac{\exp(-\phi_{\mathbf{z}}^*/\gamma)}{\sum_{\mathbf{z}' \in \mathcal{S}_{\mathbf{y}}} \exp(-\phi_{\mathbf{z}}^*/\gamma)}. \tag{7}$$

It should be noted that the entropy $\phi_{\mathbf{z}}^*$ is essentially the energy of that state $\mathbf{z}$, and the coefficient $\gamma$ plays a role of temperature in the Boltzmann distribution (LeCun et al., 2006). From this perspective, it is natural to use some classic annealing (or cooling) schedules to decrease $\gamma$ (Hajek, 1988; Nourani & Andresen, 1998; Henderson et al., 2003). In this work, we consider the following three schedules: (1) *logarithmic* cooling schedule, i.e., $\gamma_t = \gamma_0/\log(1+t)$; (2) *exponential* cooling schedule, i.e., $\gamma_t = \gamma_0 \alpha^t$; (3) *linear* cooling schedule, i.e., $\gamma_t = \gamma_0 - \alpha t$.

Furthermore, after the annealing stage, i.e., when the temperature is decreased to a small value, we will directly set $\gamma = 0$, and train the network by a few more epochs. Note that in this zero-degree stage, the problem is essentially reduced to a semi-supervised setting (Lee et al., 2013). That is, $Q_{\boldsymbol{\phi}^*}$ shrinks to a one-hot categorical distribution when $\gamma = 0$, contributing some (pseudo) labels for the learning process. Some semi-supervised techniques could be applicable to this case, but are not sufficiently efficient due to the massive state space. Therefore, we use the simplest strategy, i.e., only train by those examples with predicted symbols satisfying the symbolic constraint.

## 5 EXPERIMENTS AND RESULTS

We carry out experiments on three tasks, viz. handwritten formula evaluation (HWF), visual Sudoku classification (Sudoku), and single-destination shortest path prediction in weighted graphs (SDSP). For the proposed approach, we split it into two stages, i.e., Stage I: Annealing ($\gamma$-decreasing) stage, and Stage II: Zero-degree ($\gamma = 0$) stage, and separately evaluate Stage I and Stage I+II. For the first state, we employ three different cooling schedules (Log, Exp, and Linear) as discussed in Section 4. The projection operator is specific to each task, and the corresponding inverse projection operator is implemented by the Z3 SMT solver (Moura & Bjørner, 2008). Through parallel computation (Joblib Development Team, 2020), Z3 solves the inverse projection (on average) in 2.8ms~6.4ms per example, which is generally acceptable for batch gradient descent.

We compare our approach with the existing state-of-the-art methods, which can be divided into two categories, viz., policy-gradient-based approaches, and symbolic-parser-based approaches. The former includes RL (i.e., learning with REINFORCE) and MAPO (Liang et al., 2018) (i.e., learning by Memory Augmented Policy Optimization). For the latter, most existing methods (e.g., semantic loss (Xu et al., 2018) and deepproblog (Manhaeve et al., 2018)) are intractable in the studied

| Method | | Symbol | Calculation |
|---|---|---|---|
| | RL | 6.5 | 0.0 |
| | MAPO | 8.7 | 0.0 |
| Baseline | NGS | 8.1 | 0.0 |
| | SSL | 70.5 | 8.4 |
| | NA | 55.1 | 2.83 |
| Ours | Log | 81.4 | 23.2 |
| (Stage I) | Exp | 82.6 | 25.7 |
| | Linear | 79.9 | 19.9 |
| Ours | Log | 91.0 | 52.2 |
| (Stage I+II) | Exp | **98.6** | **90.7** |
| | Linear | 97.6 | 85.0 |

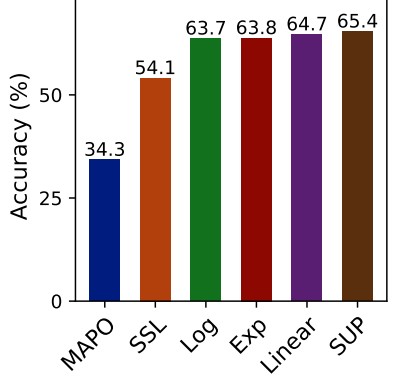

Table 1: Accuracy (%) of the HWF task. Our methods (i.e., Stage I+II) perform much better than comparison methods.

Figure 3: Accuracy (%) of the SDSP task. Our methods are better than competitors and close to the direct supervision case.

tasks (Huang et al., 2021). Hence, based on Proposition 2 and borrowing our projection-based MCMC technique, we implement a stochastic version for them (referred to as SSL henceforth) for comparison. More implementation details can be found in Appendix B. The code is available at https://github.com/SoftWiser-group/Soften-NeSy-learning.

## 5.1 Handwritten Formula Evaluation

We first evaluate our approach on the handwritten formula dataset provided by Li et al. (2020). Since the original dataset consists of formulas with lengths varying from 1 to 7, which may lead to the label leakage problem, we only take the 6K/1.2K formulas with length 7 as the training/test set. In this neuro-symbolic system, the neural network is required to recognize symbols including digits 1-9 and basic operators $(+, -, \times, \div)$. The symbolic module evaluates the expression via the Python program 'eval'. In this task, we also compare with the neural-grammar-symbolic (NGS) method (Li et al., 2020), and a special case of our approach with no-annealing strategy (NA) where we fix $\gamma = 0.001$. For SSL, NA, and our approach, we define the projection operator as $\Pi(\mathbf{z}_1; \ldots; \mathbf{z}_7) = [\mathbf{z}_1; \mathbf{z}_2; \mathbf{z}_4; \mathbf{z}_6; \mathbf{z}_7]$, i.e., drop the third and fifth symbols in the formula.[3]

We report the symbol accuracy (i.e., the percentage of symbols that are correctly predicted) and the calculation accuracy (i.e., the percentage of final results that are correctly calculated) in Table 1. Observe that our approaches (Log, Exp, and Linear) significantly outperform the competitors. Additionally, when Stage II is included, both symbol accuracy and (especially) calculation accuracy can be further improved. Overall, our two-stage algorithm with the Exp annealing strategy achieves the best performance on both symbol accuracy and calculation accuracy. The Log annealing strategy in Stage II cannot obtain a comparable result with the other two strategies, because its temperature is not reduced to a sufficiently low value. Additional learning curve results and analysis can be found in Appendix B.4.

## 5.2 Visual Sudoku Classification

We next evaluate our approach on a visual Sudoku classification task (Wang et al., 2019), where the neural network recognizes the digits (i.e., MNIST images) in the Sudoku board, and the symbolic module determines whether a solution is valid for the puzzle. To evaluate the sample efficiency of our approach, we vary the size of the training set by 50, 100, 300, and 500, and the size of the test set is fixed at 1,000. Note that the solution space in this task is intrinsically connected. For example, one can easily obtain a new solution by permuting any two digits. Therefore, we additionally include this strategy without the projection (denoted by MCMC) as a baseline.

---

[3]We observe that the solution space is well-connected through different projections. For example, for the used projection with initial $\gamma_0 = 1.0$, around 46% solutions successfully jump to other solutions in an epoch.

For a given 4-by-4 Sudoku puzzle, we divide it into four disjoint 2-by-2 subboards, and the projection drops the anti-diagonal two. In the projection space, we randomly switch two digits in different rows or columns, and the following example illustrates the whole projection process.

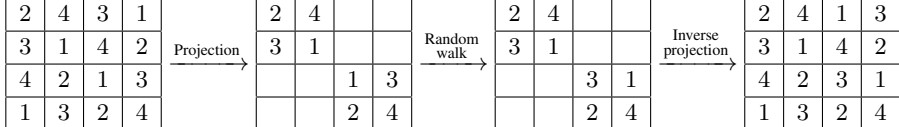

Table 2 shows the accuracy result, i.e., the percentage of correctly predicted boards. It can be seen that RL, MAPO, and SSL fail to obtain a sensible result across all cases. Although the crude MCMC method can achieve a good result, it is still significantly outperformed by our approaches. The reason is that the Markov chain obtained in the original space has a slow mixing time, making the MCMC algorithm prone to getting stuck at local minima. To investigate the grounding effect and the sample efficiency, we also report the number of training examples that are correctly grounded (i.e., satisfying the symbolic constraints) in the brackets of Table 2. The result shows the high sample efficiency of our approach. Particularly, with the number of training puzzles increased, the rate of correctly grounded examples has exceeded 90%.

Table 2: Accuracy (%) of the Sudoku task. Our methods significantly outperform the competitors.

| Method | | Number of training puzzles | | | |
|---|---|---|---|---|---|
| | | 50 | 100 | 300 | 500 |
| Baseline | RL | 0.0 | 0.0 | 0.0 | 0.0 |
| | MAPO | 0.0 | 0.0 | 0.0 | 0.0 |
| | SSL | 5.8 | 1.6 | 0.0 | 0.2 |
| | MCMC | 24.3 | 49.5 | 63.4 | 69.8 |
| Ours (Stage I) | Log | 30.7 | 74.7 | 85.9 | 86.0 |
| | Exp | 33.6 | 76.5 | 77.4 | 89.3 |
| | Linear | 46.3 | 66.8 | 79.4 | 84.1 |
| Ours (Stage I+II) | Log | 64.8 (35) | 82.2 (85) | **93.5 (279)** | 92.9 (474) |
| | Exp | 66.3 (39) | **85.5 (94)** | 92.3 (274) | **95.4 (480)** |
| | Linear | **66.9 (41)** | 81.5 (85) | 90.8 (273) | 94.0 (478) |

## 5.3 SHORTEST PATH SEARCH

We finally conduct a single-destination shortest path search task. In this neuro-symbolic system, the symbolic reasoning part implements an $A^*$ search algorithm (Russell, 2010), which maintains a priority queue of the estimated distance $d(n) = g(n) + f_{\boldsymbol{\theta}}(n)$, where $n$ is the next node on the path, $g(n)$ is the known distance from the start node to $n$, and $f_{\boldsymbol{\theta}}(n)$ is the shortest distance from $n$ to the destination heuristically predicted by a neural network. For simplicity, we set the queue length to 1, i.e., only visit the node with the shortest estimated distance. We randomly generate 3K/1K graphs as training/test set through NetworkX (Hagberg et al., 2008). In each graph, the number of vertices is fixed to 30, and the weights of edges are uniformly sampled among $\{1, 2, \ldots, 9\}$.

For this regression task, we define the symbol $\mathbf{z}$ as a multivariate Gaussian with diagonal covariance, i.e., $\mathbf{z} \sim \mathcal{N}(f_{\boldsymbol{\theta}}(\mathbf{x}), \sigma^2 \mathbf{I})$, where $f_{\boldsymbol{\theta}}(\mathbf{x})$ indicates the predicted distances from all nodes to the given destination. The dimension of $\mathbf{z}$ is 30, and the projection is defined by dropping $[\mathbf{z}_5, \mathbf{z}_{10}, \mathbf{z}_{15}, \mathbf{z}_{20}, \mathbf{z}_{25}]$. The random walk in each step selects a component of $\mathbf{z}$ and adds a uniform noise from $[-5, 5]$ on it.

Figure 3 shows the accuracy results, i.e., the percentage of shortest paths that are correctly obtained. To better understand the effectiveness, we additionally train a reference model (denoted as SUP) with directly supervised labels (i.e., the actual distance from each node to the destination). It can be observed that, our approaches not only outperform the existing competitors, but also achieve a comparable result with the directly supervised model SUP.

## 6    RELATED WORK

**Neuro-symbolic learning.** To build a robust computational model integrating concept acquisition and manipulation (Valiant, 2003), neuro-symbolic computing provides an attractive way to reconcile neural learning with logical reasoning. Numerous studies have focused on symbol grounding to enable conceptualization for neural networks. An in-depth introduction can be found in recent surveys (Marra et al., 2021; Garcez et al., 2022). According to the way the symbolic reasoning component is handled, we categorize the existing work as follows.

*Learning with logical constraints.* Methods in the first category parse the symbolic reasoning into an explicit logical constraint, and then translate the logical constraint into a differentiable loss function which is incorporated as constraints or regularizations in network training. Although several methods, e.g., Hu et al. (2016); Xu et al. (2018); Nandwani et al. (2019); Fischer et al. (2019); Hoernle et al. (2022), are proposed to deal with many different forms of logical constraints, most of them tend to avoid the symbol grounding. In other words, they often only confine the network's behavior, but do not guide the conceptualization for the network.

*Learning from symbolic reasoning.* Another way is to regard the network's output as a predicate, and then maximize the likelihood of correct symbolic reasoning (a.k.a. learning from entailment (Raedt et al., 2016, Sec. 7)). In some of these methods (Manhaeve et al., 2018; Yang et al., 2020; Pryor et al., 2022; Winters et al., 2022), the symbol grounding is often conducted in an implicit manner (as shown by Proposition 2), which limits the efficacy of network learning. Some other methods (Li et al., 2020; Dai et al., 2019) achieve an explicit symbol grounding in an abductive way, but still highly depend on a good initial model. Our proposed method falls into this category, but it not only explicitly models the symbol grounding, but also alleviates the sensitivity of the initial model by enabling the interaction between neural learning and symbolic reasoning.

*Differentiable logical reasoning.* This line of work considers emulating symbolic reasoning through a differentiable component, and embedding it into complex network architectures. To achieve this goal, a series of techniques (Trask et al., 2018; Grover et al., 2019; Wang et al., 2019; Chen et al., 2021) are proposed to approximate different modules in logical reasoning. Despite the success, these methods still succumb to the symbol grounding problem, and cannot achieve a satisfactory performance without explicit supervision (Topan et al., 2021).

**Constrained counting and sampling.** Quite a few neuro-symbolic learning methods (Manhaeve et al., 2018; Xu et al., 2018) rely on knowledge compilation (Darwiche & Marquis, 2002), which implements the exact constrained counting based on Binary Decision Diagram (BDD) (Akers, 1978) or Sentential Decision Diagram (SDD) (Darwiche, 2011). Some approximate versions (De Raedt et al., 2007; Manhaeve et al., 2021) are proposed to overcome the computational hardness (Valiant, 1979; Jerrum et al., 1986), but are still inefficient and poorly scalable to large-size problems.

Aided by the progress of SAT/SMT solving (Malik & Zhang, 2009; Vardi, 2014), randomized approximate constrained counting/sampling approaches have been proposed, which are based on hashing (e.g., Chakraborty et al. (2013); Meel et al. (2016)) or MCMC (e.g. Wei et al. (2004); Gomes et al. (2006); Ermon et al. (2012)). In particular, previous MCMC-based methods suffer from connectivity barriers. Our approach is based on MCMC, but leverages projection to overcome the connectivity barrier (Moitra, 2019; Feng et al., 2021a). Moreover, our theoretical result shows that the stochastic gradient descent can offset the possible bias of the gradient estimate introduced by MCMC and SMT solvers.

## 7    CONCLUSION

In this paper, we present a new neuro-symbolic learning framework for better integrating neural network learning and symbolic reasoning. To focus on the crucial problem of symbol grounding, we limit this work to scenarios where the symbolic reasoning logic is given as a priori knowledge. The next step is to incorporate the learning of the knowledge into our framework by, e.g., inductive logic programming. Moreover, though SMT solvers make the projection feasible in a broad range of settings, they may become a bottleneck when instantiating our framework for more complex systems. It would be interesting to consider a substitute for SMT solvers in neuro-symbolic frameworks.

ACKNOWLEDGMENT

We thank the anonymous reviewers for their insightful comments and suggestions. This work is supported by the National Natural Science Foundation of China (Grants #62025202, #62172199). T. Chen is also partially supported by Birkbeck BEI School Project (EFFECT) and an overseas grant of the State Key Laboratory of Novel Software Technology under Grant #KFKT2022A03. Xiaoxing Ma (xxm@nju.edu.cn) is the corresponding author.

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

# A Proofs

## A.1 Proof of Proposition 1

*Proof.* We define $\ell(\boldsymbol{\theta}) = \mathbb{E}_{\mathbf{z}\sim Q^*}\log P_{\boldsymbol{\theta}}(\mathbf{z}|\mathbf{x})$ as the objective function, and consider the bias of gradient on different distributions $\widehat{Q}$ and $Q^*$, which is denoted by $m(\boldsymbol{\theta})$.

$$m(\boldsymbol{\theta}) = \mathbb{E}_{\mathbf{z}\sim\widehat{Q}}\nabla\log P_{\boldsymbol{\theta}}(\mathbf{z}|\mathbf{x}) - \mathbb{E}_{\mathbf{z}\sim Q^*}\nabla\log P_{\boldsymbol{\theta}}(\mathbf{z}|\mathbf{x}) = \sum_{\mathbf{z}\in\mathcal{S}_\mathbf{y}}\left(\widehat{Q}(\mathbf{z}) - Q^*(\mathbf{z})\right)\nabla\log P_{\boldsymbol{\theta}}(\mathbf{z}|\mathbf{x}).$$

Note that our sampling strategy ensures that only feasible states will be generated, and thus we have

$$\text{supp}(\widehat{Q}) \subseteq \text{supp}(Q^*) \subseteq \mathcal{S}_\mathbf{y}.$$

Hence, through the Cauchy-Schwarz inequality, we can obtain that

$$\|m(\boldsymbol{\theta})\|^2 = \|\sum_{\mathbf{z}\in\mathcal{S}_\mathbf{y}}(\widehat{Q}(\mathbf{z}) - Q^*(\mathbf{z}))\nabla\log P_{\boldsymbol{\theta}}(\mathbf{z}|\mathbf{x})\|^2$$

$$\leq \sum_{\mathbf{z}\in\mathcal{S}_{Q^*}}(\widehat{Q}(\mathbf{z}) - Q^*(\mathbf{z}))^2 \cdot \sum_{\mathbf{z}\in\mathcal{S}_{Q^*}}\|\nabla\log P_{\boldsymbol{\theta}}(\mathbf{z}|\mathbf{x})\|^2 \leq (n\epsilon L)^2,$$

where $\mathcal{S}_{Q^*}$ represents the support of $Q^*$ and $n$ denotes its cardinal number. Now, by applying Lemma 3 in Ajalloeian & Stich (2020), we can obtain that

$$\frac{1}{K}\sum_{k=1}^K\|\nabla\ell(\boldsymbol{\theta}_k)\|^2 \leq \mathcal{O}\left(\frac{\ell\sigma^2 + \Delta_0}{\alpha\sqrt{T+1}}\right) + (n\epsilon L)^2,$$

where $\sigma^2$ is the bounded variance in gradient estimate. $\qquad\square$

## A.2 Proof of Proposition 2

*Proof.* It can be observed that the loss function $\hat{\ell}(\boldsymbol{\theta}) := -\log\sum_{\mathbf{z}\in\mathcal{S}_\mathbf{y}}P_{\boldsymbol{\theta}}(\mathbf{z}|\mathbf{x})$ is essentially the semantic loss (Xu et al., 2018, Def. 1), as well as the loss used in Deepproblog (Raedt et al., 2016, Sec. 7) and NGS (Li et al., 2020, Eq. 7).

Now, we consider the gradient $\nabla\hat{\ell}(\boldsymbol{\theta})$, which can be computed as

$$\nabla\hat{\ell}(\boldsymbol{\theta}) = -\sum_{\mathbf{z}\in\mathcal{S}(\mathbf{y})}\frac{1}{\sum_{\mathbf{z}'\in\mathcal{S}(\mathbf{y})}P_{\boldsymbol{\theta}}(\mathbf{z}'|\mathbf{x})}\nabla P_{\boldsymbol{\theta}}(\mathbf{z}|\mathbf{x}),$$

$$= -\sum_{\mathbf{z}\in\mathcal{S}(\mathbf{y})}\frac{P_{\boldsymbol{\theta}}(\mathbf{z}|\mathbf{x})}{\sum_{\mathbf{z}'\in\mathcal{S}(\mathbf{y})}P_{\boldsymbol{\theta}}(\mathbf{z}'|\mathbf{x})}\nabla\log P_{\boldsymbol{\theta}}(\mathbf{z}|\mathbf{x}),$$

Let us switch to Problem (P). By setting $\gamma = 1$, for any $\mathbf{z} \in \mathcal{S}(\mathbf{y})$, we can compute $Q_{\phi^*}$ by

$$Q_{\phi^*}(\mathbf{z}) = \frac{P_{\boldsymbol{\theta}}(\mathbf{z}|\mathbf{x})}{\sum_{\mathbf{z}'\in\mathcal{S}(\mathbf{y})}P_{\boldsymbol{\theta}}(\mathbf{z}'|\mathbf{x})}.$$

and thus $\nabla_{\boldsymbol{\theta}}\ell(\boldsymbol{\theta}, \phi^*)$ can be rewritten as

$$\nabla\ell(\boldsymbol{\theta}) = -\sum_{\mathbf{z}\in\mathcal{S}_\mathbf{y}}Q_{\phi^*}(\mathbf{z})\nabla\log P_{\boldsymbol{\theta}}(\mathbf{z}|\mathbf{x}).$$

For completeness, we also simply prove that the $Q_{\phi^*}$ is the optimal solution of $\min_\phi \ell(\boldsymbol{\theta}, \phi)$ with $\gamma = 1$. Elaborately, the Lagrangian function of the lower-level problem is

$$\mathcal{L}(\phi; \lambda) = \sum_{\mathbf{z}\in\mathcal{S}_\mathbf{y}}Q_\phi(\mathbf{z})\log\left(\frac{Q_\phi(\mathbf{z})}{P_{\boldsymbol{\theta}}(\mathbf{z})}\right) - \lambda\left(\sum_{\mathbf{z}\in\mathcal{S}_\mathbf{y}}Q_\phi(\mathbf{z}) - 1\right).$$

By computing its gradient in $Q_\phi(\mathbf{z})$, and let it vanish, then

$$\log Q_{\phi^*}(\mathbf{z}) + 1 - \log P_{\boldsymbol{\theta}}(\mathbf{z}) - \lambda = 0$$

should hold for any $\mathbf{z} \in \mathcal{S}_\mathbf{y}$. Therefore, we have

$$Q_{\boldsymbol{\phi}^*}(\mathbf{z}) = e^{\lambda-1} P_{\boldsymbol{\theta}}(\mathbf{z}), \quad \sum_{\mathbf{z} \in \mathcal{S}_\mathbf{y}} e^{\lambda-1} P_{\boldsymbol{\theta}}(\mathbf{z}) = 1.$$

Putting these two equalities together, we can obtain that

$$Q_{\boldsymbol{\phi}^*}(\mathbf{z}) = \frac{P_{\boldsymbol{\theta}}(\mathbf{z})}{\sum_{\mathbf{z}' \in \mathcal{S}_\mathbf{y}} P_{\boldsymbol{\theta}}(\mathbf{z}')},$$

which completes the proof. $\qquad\square$

## B    EXPERIMENTS

### B.1    TWO-STAGE ALGORITHM

In this subsection, we briefly discuss the proposed two-stage algorithm used in Section 5. Recall the two stages are: Stage I: Annealing ($\gamma$-decreasing) stage, and Stage II: Zero-degree ($\gamma = 0$) stage.

Stage I faithfully implements Algorithm 1. During Stage I training, with the temperature $\gamma$ decreasing, $Q_{\boldsymbol{\phi}^*}$ gradually converges to a one-hot categorical distribution, which will finally give a deterministic input-symbol mapping (i.e., a pseudo label). Ideally, if the solution space can be properly enumerated, we can start the Stage II algorithm in a fully-supervised way, i.e., fine-tuning the network by these deterministic mappings. However, the solution space is discrete and grows exponentially, and thus it is intractable to determine the mapping for each input.

To this end, we conduct Stage II in a semi-supervised way. That is, when fine-tuning the network, we only use the deterministic mappings that can be easily determined, and drop the others. Elaborately, for the given input, if the model's prediction satisfies the symbolic constraint, $Q_{\boldsymbol{\phi}^*}$ can be directly computed according to equation 5. Hence, we only use these inputs as the training data in Stage II, leading to a semi-supervised setup.

### B.2    FRAMEWORK GUIDELINE

Two key elements in our framework are annealing strategy and projection operator. Hence, we briefly discuss how to set the temperature in the annealing strategy and the projection operator.

(1) **The setting of temperature.** Intuitively, a good initial temperature should ensure the new state will not be rejected at the first few training epochs. Therefore, setting the initial temperature to a large value (e.g., $\gamma_0 = 1$ in our three tasks) is generally effective. For the hyperparameter setting in the annealing strategy, we recommend to follow that of Nourani & Andresen (1998).

(2) **The selection of projection operator.** The selection of variables to be dropped by the projection operator is very critical in our framework. Feng et al. (2021a) propose to evaluate the quality of projection operator via entropy, which hints at choosing the variables with less entropy decreasing. A more direct and practical guideline is to drop variables that are highly correlated to others, because these variables depend on others and thus have lower entropy.

(3) **The setting of projection dimension.** The dimension of the projection space $\Omega$ requires a trade-off: a larger $\dim(\Omega)$ cannot effectively improve the connectivity of solutions, while a smaller $\dim(\Omega)$ may introduce more bias by the SMT solver. A practical method to determine $\dim(\Omega)$ may be via trail-and-error, i.e., to gradually decrease the dimension of the projection space until the connectivity of $\Omega$ is satisfactory. Furthermore, there are different methods which may be used to measure the connectivity of the solution space. In theory, one may count the number of connected components of the solution space, which is not very practical. In our experiments, as we carry out random walk, we adopt the number of random walk steps needed for the transition from the initial solution to a target solution. For example, in the HWF task, $\dim(\Omega)$ is set as 5, since we observe that the solutions are fully connected by dropping the third and fifth symbols.

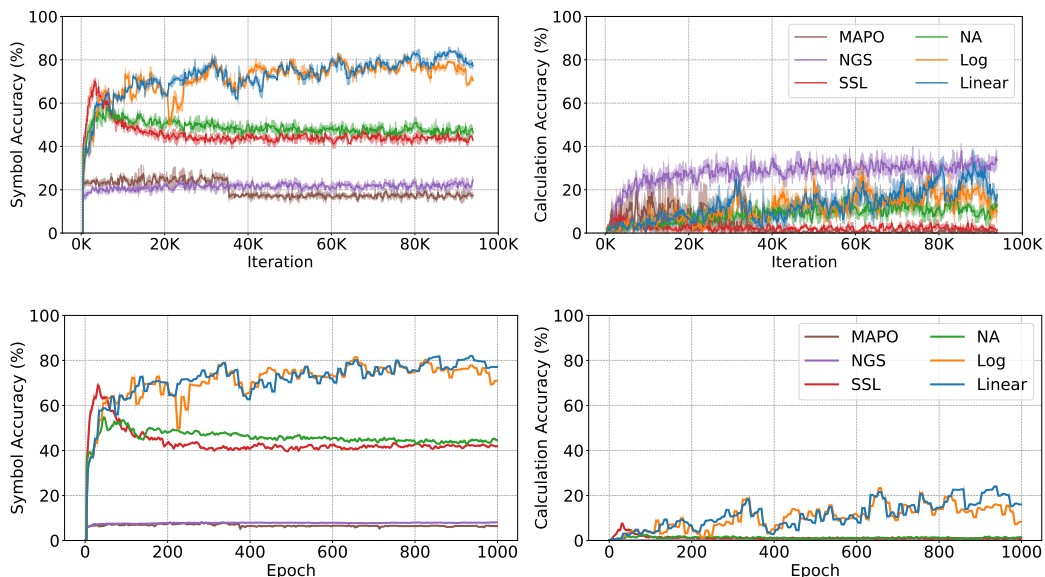

Figure 4: Training curves (the first row) and test curves (the second row) of different approaches. We only plot the curves for some of the methods for brevity. Our approaches (Log and Linear) achieve the best symbol accuracy on the training set, and also generalize better to the test set.

### B.3    EXPERIMENTAL SETTING

**Model architecture.** For HWF and Sudoku tasks, we used the LeNet-5 architecture; For SDSP task, we used the multilayer perceptron with $30 \times 30$ input neurons, one hidden layers with 128 neurons, and an output layer of 30 neurons.

**Batch size and epoch.** For all tasks, the batch size was set to 64. For all comparison methods and our Stage I algorithm, the number of epochs is fixed to 1,000. For our Stage II algorithm, the number of epochs is fixed at 30. We fix $T = 10$ in Alg. 1, i.e., conducting ten random walk steps before one gradient descent step.

**Gradient descent algorithm.** For all comparison methods, we followed the learning algorithm setting in their respective Github repository. To be specific, RL, MAPO, and SSL conducted the Adam algorithm with learning rate 5e-4. For our approaches, we used the SGD algorithm with learning rate 0.1 in Stage I, and the Adam algorithm with learning rate 1e-3.

**Implementation.** For RL, MAPO, and NGS methods, we used the code provided by NGS authors. For SSL and NA methods, we implemented them with the same projection technique and random walk strategy with our approach. The temperature $\gamma$ is fixed to 0.001 in the NA method.

### B.4    ADDITIONAL RESULTS

For the HWF task, we plot the training/test curves of our Stage I algorithms (Log and Linear) and comparison methods (MAPO, NGS, SSL, and NA) in Figure 4. For our approaches, the random walk step is also counted within the iteration. First, it can be observed that the policy-gradient-based method (MAPO) cannot well fit the training data due to the issue of sparse reward. For the NGS method, it quickly overfits the training set, but cannot improving the symbol accuracy and generalizing to the test set. This result is not surprising because the back-search in NGS is too greedy and hence only works with a good initial model. SSL and NA can be treated as two different variants of our framework, and they learn well during the first few epochs, but then collapse due to the lack of an effective grounding.

In Table 3, we further report some results of additional experiments on the HWF task. We consider different combinations of our method with the existing methods. We first apply the Stage II

Table 3: Results (%) of additional experiments.

| Method | Symbol | Calculation |
|---|---|---|
| SSL + Stage II | 17.4 | 1.41 |
| NA + Stage II | 13.0 | 0.41 |
| Stage I + NGS | 99.5 | 96.6 |

algorithm for SSL and NA. However, such variants collapse since they cannot provide a sufficient calculation accuracy, and finally converge to nearly zero calculation accuracy. We next apply the back-search in NGS after our Stage I algorithm, by initializing with our Stage I models. This variant can achieve comparable results with that using direct supervision. Note that a bit accuracy drop compared with that in the original NGS paper is due to that we only evaluate the model on length-7 formulas. This result further verifies the effectiveness of our softened symbol grounding. However, the back-search in NGS lacks versatility in more complex settings and is not applicable to other studied tasks.

