# OpenReview forum: "Softened Symbol Grounding for Neuro-symbolic Systems"
_ICLR.cc/2023/Conference — ICLR 2023 poster_

### Official Review · Reviewer_ypT3 · 2022-10-20

**Confidence:** 2
**Correctness:** 2
**Technical Novelty And Significance:** 2
**Empirical Novelty And Significance:** 1
**Recommendation:** 5

**Clarity, Quality, Novelty And Reproducibility:**

As I wrote in the previous section, the paper is unreadable for readers outside of this subfield, and I believe that this complication is not needed, and could have been written in a more friendly way.
For example:

* The abstract was completely unreadable for me. Although I understood every individual word, when reading the abstract, I could not understand anything from the contributions:
```
(1) modeling of deterministic symbol solution states as a Boltzmann distribution, which avoids expensive state searching and facilitates the interaction between network training and symbolic reasoning;
(2) an efficient
MCMC sampling technique leveraging projection and SMT solvers, which overcomes the connectivity barrier in sampling symbol solution spaces;
(3) an annealing mechanism that avoids the trap of sub-optimal symbol groundings
```
(what are `deterministic symbol solution states`? What is the `connectivity barrier`? what is `an annealing mechanism`? What is `the trap of sub-optimal symbol groundings`?)

* The introduction continues to be completely unreadable, for example:
```
From the perspective of game theory, the softening forms a series of mixed strategy games during the annealing process, which encourages  stronger interactions and thus obtains a strict improvement of the utility (Conitzer, 2016).

Furthermore, the softening approach allows us to solve the original bi-level optimization by the (min-oracle) stochastic gradient descent (Jin et al., 2020). To avoid the sparsity problem and obtain the gradient estimate efficiently
```
and so on. (how is this related to "mixed strategy games", and what are "mixed strategy games"? What is this "softening approach"? what is exactly the "bi-level optimization" problem? What is "min-oracle"?)

* Later in Section 2:
```
Advantages. Game theory provides a perspective to understand why our softened strategy improves over problems (1) and (2) with a better interaction between model training and symbolic reasoning. Either problem (1) or (2) can be viewed as a Stackelberg game with pure strategies, in which the player takes a single action (i.e., selecting a deterministic mapping h). However, in problem (P), the player takes a randomized action with the distribution Qϕ. Such a mixed strategy does provide more information of the game, and thus strictly improves the utility (Letchford et al., 2014).
```
What is a "Stackelberg game", and how does it "improve the utility" (and what is exactly the utility here)? Such paragraphs just makes it frustrating for the reader.

* `By applying the Metropolis algorithm` ... `creating the so-called connectivity barrier.` What is the Metropolis algorithm? What is the so-called connectivity barrier? Readers that are not familiar with this algorithm and barrier cannot follow the paper when such terms are mentioned.

## Additional comments
* I could not estimate whether the baselines are sufficient and whether the approach is novel.
* Presentation - the authors have modified the `hyperref` package settings in a way that does not highlight the hyperlinks and makes it difficult to read. Why? Please fix that.

**Strength And Weaknesses:**

## Strengths
* The problem is interesting and important, as providing good solutions will allow neural models to leverage existing and exact tools (e.g., a calculator) instead of learning to calculate themselves, since a calculation by the neural network is expected to make mistakes, and is known to not generalize well.
* The proposed technique sounds correct and clever, although I did not understand most of the details.
* The approach is evaluated on three tasks and shows significant improvements over the baselines.

## Weaknesses
* Readability - I did not understand most of the technical details, and the paper is not very friendly for readers out of this sub-field.
I did understand the general idea, but many mathematical steps are mentioned only briefly or cite a reference, which makes the reader feel that they are expected to read many references before being able to understand the paper.
I do believe that the paper could have been written in a way that will make it more readable to the broad ICLR audience.
See details in the next section.


**Summary Of The Paper:**

The paper addresses the general family of tasks when the input needs to be processed by a neural model, then processed by a symbolic model, and produce an output.
For example, the input may be an *image of* a handwritten arithmetic calculation, e.g. `7*8-5-9`. The neural model needs to recognize the digits and the operators, and is then allowed to use a symbolic model such as a standard simple calculator to produce the result (`42`).
These tasks are very difficult when we only have labels for the final answer, i.e. `42`, and no intermediate labels for the digits themselves.

The paper proposed a "soft" way to solve this problem by introducing a probability over the input, and a way to solve and optimize.

**Summary Of The Review:**

I appreciate the difficulty of the problem that this paper is trying to solve, and the empirical results look promising.
However, I could not understand most of the paper,  which sometimes seems to be written cryptically deliberately.

I thus vote for rejection with low confidence, because even if the paper's solution is sound and novel, the paper will be valuable only to a very limited audience.

---

> ### Author Response · Authors · 2022-11-14
> **Response to Reviewer ypT3**
>
> Thanks for urging us to improve readability. We have revised the paper to make it more friendly to the board ICLR audience. Particularly, we extensively rewrote the introduction, and re-drew Figure 1 to illustrate the challenge of symbol grounding in neural-symbolic learning and the basic idea of our proposed framework. We also avoided some non-essential technical terms (e.g., min-oracle and bi-level problem). For those at the core of our approach, namely, Game Theory concepts such as *mixed-strategy game* and *Stackelberg game* and MCMC Sampling concepts such as *rapid mixing* and *Metropolis algorithm*, we gave more explanations and/or references in the context.
>
> About the hyperlink: we had not modified the hyperref package settings. Is it due to viewing the pdf with the embedded viewer in a browser? We find that the hyperlinks are highlighted in Preview but not in Chrome.

---

> > ### Comment · Reviewer_ypT3 · 2022-11-18
> > **Thank you**
> >
> > Thank you for revising the paper.
> >
> > To me, the paper is still unreadable.
> > For example, the Metropolis algorithm is mentioned with citations now on page 4, but not explained even briefly. These citations do not really help me, because a reader will not read the references to understand this one detail.
> >
> > Similarly, the Stackelberg game was not explained as well. A reader who has never heard about this term will not be able to continue reading the paper after the point where it is mentioned.
> >
> > Maybe the paper is good and useful for people in its subfield, but for me, it is not standalone and unreadable, and I thus keep my score.

---

### Official Review · Reviewer_78tc · 2022-10-22

**Confidence:** 3
**Clarity, Quality, Novelty And Reproducibility:** Please see strengths and weaknesses a…
**Correctness:** 4
**Technical Novelty And Significance:** 3
**Empirical Novelty And Significance:** Not applicable
**Recommendation:** 6

**Strength And Weaknesses:**

Strengths:
* The paper presents a reasonably straightforward but well motivated idea for allowing the perceptual and symbolic reasoning aspects of a neuro-symbolic method to interact. While I am not an expert in this kind of neurosymbolic reasoning, the method was well motivated by gaps in related work as pointed to.

* The method is at least partially theoretically justified, including the use of stochastic gradient descent as a method for removing some of the sampling bias introduced by the paper’s MCMC procedure.

* Neurosymbolic models have seen increasing attention over the last several years, and given this paper’s strong results, it seems well posed to have an impact in the community.

Weaknesses:
* Baselines seem especially weak and oddly chosen given the extensively cited related work that is more recent. For example, Li et al, 2020, which seems like one of the most recent and related pieces of work, is only compared on the first of the three tasks, without explanation for why it is not used for the other two.
* The reasoning for the variants of the chosen tasks was confusing and seemed misleading. For example, the paper states “Since the original dataset (Li et al, 2020) consists of formulas with lengths varying from 1 to 7, which may lead to the label leakage problem, we only take the 6K/1.2K formulas with length 7 as the training/test set.” What does this mean? What would happen if you compared on all formulas? This seems particularly important because the performance of Li et al in this paper is *90 absolute percentage points* worse than the original paper. To ensure there isn’t a bug with the implementation, or the interpretation of the experimental results, I would like to see a comparison to their method on the full dataset as opposed to just the length 7 formulas. I also do not understand how including the whole dataset would lead to a label leakage problem.
* The paper has several issues with clarity that would need to be improved before it could be published.
  * For example, the authors use “Stage I” and “Stage II” to describe different aspects of their method throughout the results, but Stage I and Stage II are never defined in the text, so I was not able to follow what this was supposed to mean.
In proposition 2, the paper states that the method is a stict generalization of prior work which is the special case of gamma = 1. I believe this should be gamma = 0 (the deterministic case).
  * Similarly there is jargon in several places that general machine learning readers may be unfamiliar with, for example “Either problem (1) or (2) can be viewed as a Stackelberg game with pure strategies, in which the player takes a single action (i.e., selecting a deterministic mapping h)”. What is a Stackelberg game? Why is this important?
  * I did not understand the point about how after the annealing stage the problem becomes a semi-supervised one. Perhaps this could be spelled out a bit more clearly with some examples of what the semi-supervised problem looks like (what is the data, what parts of it are unlabeled, what are the labels).
* The paper is missing key ablations to support the claims.
   * In particular, a main claim is that using an annealed gamma for first having stochastic representations of symbols and then sharpening them over time is not directly ablated. It would be nice to see a comparison to the method when gamma is always fixed to be 0 from the start of training, as this is the key difference to prior work (as also noted by the paper’s proposition 2 remarks).
   * A secondary claim relates to the projection operator, and its importance for enabling well-behaved MCMC proposals. The projection operator is chosen in a bespoke way for each problem. How sensitive is the method to the particular choice of projection operator? What happens if no projection operator is used? Given that this choice injects a significant amount of domain knowledge from a practitioner and does not appear to generalize across problems, this seems like an important experiment to run.





**Summary Of The Paper:**

This paper presents a neurosymbolic learning approach for tackling the problem of symbol grounding when the logic of symbolic constraints is provided.Their method uses an SMT solver to initialize one possible symbol grounding for a problem such that it satisfies a given “solution constraint”, and then uses a Boltzmann machine to enable sampling from a distribution over potential symbol groundings that satisfy a given solution constraint (such as a visusal equation taking on a particular value). The method is evaluated on three tasks: visual math, visual sudoku, and visual shortest path finding. The method is compared to baselines from reinforcement learning and symbolic-parser-based approaches. Across the tasks, they find that their method significantly outperforms the baselines.

**Summary Of The Review:**

Despite the novel and interesting method introduced in this paper, there are major weaknesses in the clarity and quality of the paper, specifically as it relates to missing ablations, sensitivity analyses, and justification for task modifications. If the concerns can be addressed by the authors, and the paper’s clarity improved, I am willing to increase my score.

---

> ### Author Response · Authors · 2022-11-14
> **Response to Reviewer 78tc**
>
> Thanks for the comment.
>
> **1. Baselines**: we compare NGS (Li et al, 2020) only in the first task because the back-search in NGS lacks versatility in more complex settings and is not applicable to the other studied tasks. We will highlight this in the revision.
>
> **2. Label leakage in  (Li et al, 2020)**:
>   - A length-1 formula is actually a single digit (e.g., "1"=1 and "2"=2), and thus incorporating them into the training data means showing the model the intermediate labels of the latent symbols. Similarly, length-3 formulas also have this issue. For example, given the calculation result 4.5, the hidden symbol can be easily decided (i.e., only "9/2"=4.5). Therefore, in our experiment, we choose the hardest case: using only length-7 formulas to show the effectiveness of our approach.
>   - For the NGS (Li et al, 2020) method, we used the code provided by the authors. The ineffectiveness of NGS in the length-7 case is because NGS cannot successfully find a feasible state (the state space is much larger in the length-7 case). Moreover, in Appendix B.3, we showed that NGS can achieve good performance (96.6% accuracy), if it uses our Stage I model as its initial model.
>   - In fact, as shown by our experiment, our method with  length-7 formulas only is still comparable to NGS with the full set of formulas.
>
> **3. Clarification**:
>   - Stage I and Stage II: Stage I and Stage II are introduced in the first paragraph of Section 5, and we have highlighted it in the revision. We have also added a discussion about the two stages in Appendix B.1.
>   - $\gamma$ in Proposition 2: we believe that this is a misunderstanding.  Please refer to a detailed proof in Appendix A.2.
>   - Stackelberg game: we have revised the sentence for better readability. This is a notion in game theory. In the Stackelberg game, one player moves first and the other moves afterwards. This provides insight into why the proposed approach is superior to the alternatives.
>   - Semi-supervision: the annealing stage decreases $\gamma$ to zero, and now $Q_{{\phi}}$ is reduced to a one-hot categorical distribution, i.e., $Q_{{\phi}}$ indicates a precise label. If the derived label satisfies the symbolic constraint, we will train the network with it, which essentially is in a semi-supervised setting. We have added the related discussion in Appendix B.1.
>
> **4. Ablation study**:  Actually the required ablation study was included in the original experiments. We have revised the paper to make it clear.
>   - Fixing $\gamma=0$: This variant was carried out as a special case of our approach with no-annealing strategy in the first task, and reported as the NA method in Table 1. The $\gamma$ is fixed to a very small constant $0.001$ (this setting is more stable and numerically equivalent to $\gamma=0$), and we can see that this variant is far less effective.
>   - Projection operator: we have included the result when no projection is used in the second task (cf. "we additionally include this strategy without the projection (denoted by MCMC) as a baseline" in the first paragraph of Section 5.2). From the results in Table 2, we can see that this variant is much less effective.  As to the choice of projection operators, we agree that this may need some domain expertise, but we provide some (general) guidelines in Appendix B.2 for projection operations.

---

> > ### Comment · Reviewer_78tc · 2022-11-20
> > **Thanks for the clarifications**
> >
> > Thank you for the clarifications and for pointing to the relevant ablations / experiments. Thank you also for making some of the wording more explicit, it's much easier to track if "Stage I" and "Stage II" are explicitly rather than implicitly defined.
> >
> > I still think that the overall exposition and method explanation are very difficult to parse for a reader not already familiar with this area. However, I think the experiments are strong and the scientific contribution seems solid, which is why I am increasing my score.

---

### Official Review · Reviewer_qkDm · 2022-10-23

**Confidence:** 3
**Correctness:** 3
**Technical Novelty And Significance:** 3
**Empirical Novelty And Significance:** 3
**Recommendation:** 8

**Clarity, Quality, Novelty And Reproducibility:**

The paper is clearly written, of high quality and original to the best of my knowledge.

**Strength And Weaknesses:**

As the authors mention, the key problem in the neural-symbolic systems is how to efficiently search for the feasible hidden symbol state in the exponentially large and discrete space. The authors propose a very interesting formulation to address the problem, prove the convergence and even connect with existing works. The entropy formulation that softens symbol grounding is really interesting and I quite like the accompanied sampling method proposed for the connectivity barrier. Both empirical results and the theoretical analysis are convincing enough.

I do not have many technical questions but would rather like to ask some open ones.

To me, the scope of problems that this method can be applied is limited, a minor problem though. The problems studied in this work are only challenging without hidden symbol supervision. However, in practice, for the toy problems studied, it is not hard to obtain, and may take only a few hours to get enough data to train a decent symbol grounding network. Is it possible to show your method applied to problems where data is either too complex to label for humans while your method can obtain good performance, or it simply is too bordering to get labels for intermediate representation?

Another issue with the method is efficiency. I can see that the sampling process, though improved, could still be very slow (or correct me if not). SMT solvers could also be a major bottleneck for efficiency, as they might only be feasible for moderately small problem scales. Can you show some quantitative results regarding efficiency?

**Summary Of The Paper:**

The paper proposes softening the symbol grounding problem in joint neural-symbolic learning. The major challenge in the neural-symbolic learning system studied in this work is the lack of supervision for the symbol grounding problem. This work presents efforts to model the symbol grounding problem as a Boltzmann distribution with efficient MCMC sampling to bridge the worlds of neural grounding and symbolic reasoning. Of particular interest is the sampling method the authors use to resolves the connectivity barrier in searching for feasible symbol representation. In experiments involving handwritten formulae, visual sudoku, and shortest path, the method shows improved performance compared to existing baselines.

**Summary Of The Review:**

The paper is well done and the strengths far outweigh the weaknesses.

---

> ### Author Response · Authors · 2022-11-14
> **Response to Reviewer qkDm**
>
> Thanks for the comments. We are happy to discuss these open questions.
>
> **Motivation of weakly supervised setting**. We agree that for these problems it would be possible to walk around weak supervison with some effort of symbol labeling. However, there are other motivations apart from saving labeling cost for solving weakly supervised problems with neuro-symbolic learning based on proper symbol grounding:
> - Direct labels can overly specify. It is interesting to observe that, in the handwritten formula evaluation problem our approach learned the exact label for each digit, but in the modified Sudoku problem our approach gave any of *the permutations of 1, 2, 3, and 4* for the 4 different glyph images. This difference is exactly what is needed by the different logic constraints and training data of the two problems (think of an alien who plays Sudoku but never learns Arabic numerals).
> - Neuro-symbolic learning potentially has better robustness and generalizability. The reason is that, the logical knowledge is more stable, accurate, and generalizable than labels. We conduct an additional experiment to support this point. Specifically, we inject 10% noisy data (with incorrect labels) into the handwritten formula dataset, and compare our neuro-symbolic framework with fully supervised learning. The result shows that the label noise downgrades the accuracy of full supervised learning from 95.3% to 84.5%, whileit is only decreased by 0.5% (from 90.7% to 90.2%) for our framework.
> - We believe weak supervision naturally resembles how human intelligence works, where intuitive perception and logical reasoning are integrated and bridged by symbol grounding [1].
>
> **Training efficiency of our method**. The sampling method in our framework is efficient in both theory and practice. Theoretically, with a good projection, MCMC has a rapid mixing time. Practically, we only conduct ten random walk steps in each epoch, and it is sufficient to converge. Nevertheless, the SMT solver is indeed a potential bottleneck, although modern SMT solvers such as Z3 is effective and efficient in a wide range of real-world applications (including automated theorem proving, program analysis/verification, and software testing. E.g., they are used to flush out design errors in the logical functioning of modern digital electronic chips, deployed in, e.g.,  Intel and Arms).  Also note that, besides using SMT solvers as general tools for inverse projection, one can also leverage more efficient domain-specific solvers. For example, in the Sudoku task, one can use an integer linear program solver (e.g., Gurobi). We empirically observe that the Gurobi solver could boost the efficiency by at least 100x over the SMT solver. In addition, some techniques (e.g., [2]) to improve the efficiency of SMT solvers may also be helpful in our framework.
>
> [1] S. Harnad, “The symbol grounding problem,” Physica D: Nonlinear Phenomena, vol. 42, no. 1, pp. 335– 346, 1990.
> [2]  Balunovic, M., Bielik, P., & Vechev, M. (2018). Learning to solve SMT formulas. Advances in Neural Information Processing Systems.

---

> > ### Comment · Reviewer_qkDm · 2022-12-08
> > **Reply**
> >
> > Thanks for the clarification. I believe the discussion could be integrated into the paper to make it more comprehensive. In general, I think this is good work.

---

### Official Review · Reviewer_262g · 2022-11-04

**Confidence:** 5
**Correctness:** 4
**Technical Novelty And Significance:** 4
**Empirical Novelty And Significance:** 3
**Recommendation:** 10

**Clarity, Quality, Novelty And Reproducibility:**

Each individual technique used in this paper is standard, but the particular combination has great novelty and elegance, especially the interaction between MCMC sampling and SMT solving. The symbol grounding problem studied in this paper is fundamental, and the result
achieved in this paper significantly outperforms several state-of-the-art approaches, which are highly non-trivial. Detailed implementations and evaluation datasets have been shared publicly, so the results should be easily reproducible.



**Strength And Weaknesses:**

Strengths:
- the paper is very well-written; all necessary background  (either classic or recent work) is carefully introduced, and related work is organized in a systematic and insightful manner.
- the paper addresses a fundamental problem in AI from a novel and promising angle -- view symbol grounding distribution as a mixed strategy in game playing; the combination of the MCMC sampling with SMT solving is both novel and elegant
- the paper has extensive experimental evaluations and achieves superior performance over many state-of-the-art approaches
- the paper also presents a formal analysis of the convergence guarantee (with some mild assumptions)

Weaknesses:
- the projection and inverse operations may be quite sensitive and require some non-trivial domain expertise
- the visual Sudoku classification task seems to be a simplified version of the original one. The 4-by-4 Sudoku puzzle used in the evaluation is much simpler than a standard 9-by-9 Sudoku. SATNet should be the state-of-the-art for the Sudoku task, which is however not included



**Summary Of The Paper:**

This paper presents a neuro-symbolic learning framework with an explicit design for addressing the symbol grounding problem.  The key idea is softening symbol grounding by using a Boltzmann distribution to represent the entire symbol space, rather than a specific symbol grounding. Such a design achieves an efficient interaction between neural perception and symbolic reasoning. Then, a novel MCMC sampling method combined with SMT solving is proposed to facilitate learning the correct symbol grounding. Furthermore, an annealing mechanism with three different realizations is applied to avoid sub-optimal symbol groundings. Experimental evaluations show significant improvement over several state-of-the-art approaches.


**Summary Of The Review:**

This paper makes significant progress toward addressing a fundamental problem in neuro-symbolic AI. The presented approach is novel, effective, and elegant, which is likely to have a great influence on future neuro-symbolic system design.

---

> ### Author Response · Authors · 2022-11-14
> **Response to Reviewer 262g**
>
> Thanks for the comments.
>
> **Projection operation**: We agree that in our approach the efficiency of sampling is sensitive to the projection operation. We have provided a guideline in Appendix B.2 for projection operation design. It may require domain knowledge and experience to design an effective and efficient projection function in general, although it appears to be straightforward in tasks such as the handwritten formula evaluation and shortest path searching. We believe the automated selection of projections is possible (e.g., based on entropy measurement), but that would be a topic for future work.
>
> **Sudoku with SATNet**: We did not compare with SATNet on Sudoku because that would be unfair for SATNet. Our work focuses on symbol grounding and assumes that the logic constraint is explicitly given, while SATNet tries to implicitly learn the logic constraint. The Sudoku task, in the form of what was used to evaluate SATNet, would be trivial for us because the SMT solver would easily find the solution.
>
> &emsp; We can reformulate the problem to fit our purpose, i.e., using 9-by-9 puzzle image as the input, and whether the input puzzle is solvable as the label (in contrast to giving the solution itself). Without much optimization, the preliminary results show that our method can achieve a satisfactory accuracy of 87.2%. SATNet does not seem to work in such a weakly supervised task.
>
> &emsp; In summary, our approach and SATNet are not directly comparable. Our approach builds on explicitly given logic constraint but can work with very weak supervision, while SATNet does not need explicit logic constraint but requires much stronger supervision, and most likely a good initial model [1].
>
> [1] Topan, S., Rolnick, D., & Si, X. (2021). Techniques for Symbol Grounding with SATNet. Advances in Neural Information Processing Systems.

---

### Decision · Program_Chairs · 2023-01-20

**Decision:**

Accept: poster

**Justification For Why Not Higher Score:**

This paper was ranked very highly by two reviewers, but two other reviewers had trouble understanding it.

The paper makes *very* strong assumptions about the problem class, hence addressing a relatively narrow  subcategory of neuro-symbolic problems.  That's okay, but the paper doesn't describe at all how the particular problem they're addressing relates to a broader class of problems, and it takes for granted that the subproblem is interesting.

To have a spotlight or oral, the messaging should be clearer.

**Justification For Why Not Lower Score:**

The technical strategy in the paper is interesting and might have applications beyond this particular narrow problem class.

Two expert reviewers were very positive.

The authors' answer to the reviewer's question, providing motivation for the semi-supervised setting was good.

**Metareview: Summary, Strengths And Weaknesses:**

This paper addresses the problem of "symbol grounding" for systems that have an architecture in which there is a neural network that maps an image-based input into a symbolic representation of the problem and then a fixed symbolic "reasoning" module that outputs the solution to the problem.  The system has only "end-to-end" supervision and so a primary problem is coming up with a symbolic interpretation of the input signal, such that when it is fed into the reasoner, the observed desired output is obtained.

Strengths:
- A good technical solution to a problem of interest

Weaknesses:
- The paper takes a relatively narrow view of what constitutes a "neuro-symbolic" system and (especially in the section on "Generalization of existing methods" but also rhetorically earlier) implies that *all* neurosymbolic methods are a special case of this one.   But (even in surveys by Garcez, Lamb, etc.) it is clear that there is an enormous variety of possible interpretations of "neurosymbolic" and different problem classes, solution types, etc.   It will be very important to make clear in the introduction and abstract (ideally even in the title) that this paper addresses a particular form of NS system, in which the input is an instance of a known type of symbolic reasoning problem,  presented in the form of a continuous signal (e.g. image) and the primary difficulty is arriving at a symbolic interpretation of the input.

Advice:
- Please try to make the related work section more informative.  Help the reader *really* understand what is different between your approaches and theirs.  Statements like "X cannot achieve satisfactory performance" are not helpful.  Why are they not satisfactory?  Why is it that your method has better performance?
- Some small errors: "to easy presentation";  Starting paragraph with "Elaborately" should probably be "To elaborate," or "More precisely".

**Note From Pc:**

if the above contains the word "oral" or "spotlight" please see: "oral" presentation means -> notable-top-5% and "spotlight" means -> notable-top-25%. As stated in our emails, we are disassociating presentation type from AC recommendations